# SCORE AUGMENTATION FOR DIFFUSION MODELS

## ABSTRACT

Diffusion models have recently achieved remarkable advances in generative modeling, yet we show that they are still prone to overfitting, especially when trained with limited data. To address this issue, we introduce Score Augmentation (ScoreAug), a data augmentation framework tailored for training diffusion models. Unlike conventional methods that augment clean data, ScoreAug operates directly on noisy data, naturally aligning with the denoising process of diffusion models. Moreover, the denoiser is required to predict the transformed target of the original signal, establishing an equivariant learning objective. This equivariance enables learning of scores across diverse denoising spaces – a principle we call score augmentation. We provide theoretical analysis of score consistency under general transformations, and empirically validate ScoreAug across CIFAR-10, FFHQ, AFHQv2, and ImageNet, with U-Net and DiT backbones. Results show consistent performance improvements over baselines, effective mitigation of overfitting under varying data scales and model capacities, and stable convergence. Beyond improved generalization, ScoreAug avoids potential data leakage in certain scenarios and can be seamlessly combined with standard augmentation strategies for further gains.

## 1 INTRODUCTION

Diffusion models (Ho et al., 2020; Song & Ermon, 2019) have rapidly emerged as a leading paradigm in generative modeling, demonstrating state-of-the-art performance across a wide range of real-world applications such as image generation (Dhariwal & Nichol, 2021; Rombach et al., 2022), video generation (Ho et al., 2022), and text generation (Nie et al., 2025). Unlike previous advanced methods, such as generative adversarial networks (Goodfellow et al., 2014; Hou et al., 2022), which rely on adversarial optimization, diffusion models are trained through iterative denoising with a simple and stable objective. This learning paradigm not only ensures robust optimization but also enables strong data-fitting capabilities. Recent work shows that diffusion models trained on the same dataset often converge to nearly indistinguishable score functions, producing highly similar outputs (Gu et al., 2023). While this highlights their remarkable ability to approximate data distributions, it also raises a critical concern: the very same capacity that ensures accurate modeling may exacerbate the risk of overfitting, especially in data-limited regimes.

Despite their widespread adoption, overfitting in diffusion models remains relatively underexplored. Our empirical study reveals that diffusion models can severely overfit when trained with limited data or excessive model capacity. Conventional regularization strategies, such as dropout (Srivastava et al., 2014) or weight decay (Krogh & Hertz, 1991; Loshchilov & Hutter, 2017), alleviate the problem only partially and often at the expense of generative quality. Data augmentation (Cubuk et al., 2018), long established as a key technique for improving generalization in discriminative models, has also been adapted to generative modeling (Jun et al., 2020; Zhao et al., 2020; Hou et al., 2023). However, existing augmentation methods typically operate on clean data, overlooking the fact that diffusion training fundamentally involves denoising noisy inputs (Karras et al., 2022; Ho et al., 2020; Song & Ermon, 2019). Furthermore, naive application of heuristic augmentations may introduce distribution mismatches, requiring additional conditioning mechanisms to avoid artifacts. These limitations highlight the need for an augmentation strategy that is both principled and aligned with the unique training dynamics of diffusion models.

In this work, we propose Score Augmentation (ScoreAug), a novel augmentation framework designed specifically for diffusion models to address the aforementioned issue. Unlike standard approaches that only transform clean data, ScoreAug applies augmentations directly to noisy inputs. At the same time,

the denoiser is trained to predict the transformed target, establishing an equivariant learning signal that tightly integrates with the diffusion process. This design not only enables the denoiser to learn score functions across multiple transformation spaces but also naturally mitigates data leakage risks that may arise from noise invariance. Our theoretical analysis further characterizes the relationships between score functions under general transformations, providing a principled foundation for the method. Empirical results on CIFAR-10, FFHQ, AFHQv2, and ImageNet with both UNet (Karras et al., 2022) and DiT (Peebles & Xie, 2023; Ma et al., 2024) architectures demonstrate substantial improvements in generation quality, robustness against overfitting, and compatibility with existing augmentation techniques.

## 2 RELATED WORK

### 2.1 DIFFUSION MODELS

Diffusion models have recently emerged as a powerful framework for generative modeling. The foundational work of DDPM (Ho et al., 2020; Sohl-Dickstein et al., 2015) established discrete-time diffusion processes with variational training, while DDIM (Song et al., 2020a) introduced deterministic and fast sampling through non-Markovian trajectories. On the other hand, NCSN (Song & Ermon, 2019) proposed noise conditional score matching to learn the Stein score (Liu et al., 2016) of perturbed data distribution at multiple noise levels. Annealed Langevin dynamics (Roberts & Tweedie, 1996) is then used to sample from the noise (Song & Ermon, 2020). A unifying perspective emerged through stochastic differential equations (SDEs) (Song et al., 2020b), which generalized DDPM and NCSN to continuous-time dynamics. The SDE framework categorizes diffusion processes into variance preserving (VP), variance exploding (VE) formulations, and sub-VP. Subsequent works such as flow matching (Lipman et al., 2022) and rectified flow (Liu et al., 2022) can be seen as the sub-VP form. EDM (Karras et al., 2022) further unified the formulations of VP, VE and sub-VP under a single training framework with disentangled sampling parameters, later refined in EDM2 (Karras et al., 2024) for enhanced training dynamics. On the variational perspective, VDM (Kingma et al., 2021) and its improved variant VDM++ (Kingma & Gao, 2023) established theoretical connections to maximum likelihood estimation. Recent breakthroughs in DiTs (Peebles & Xie, 2023; Ma et al., 2024) demonstrate how transformer architectures can replace traditional U-Net backbones (Rombach et al., 2022), achieving state-of-the-art results (Esser et al., 2024). In terms of generalizability of diffusion models, recent works (Somepalli et al., 2023; Carlini et al., 2023) reveal that diffusion models tend to memorize training data when model capacity exceeds dataset size, raising concerns about replication risks. Theoretical analyses (Li et al., 2023) establish polynomial relationships between generalization error bounds and sample size and model capacity. Complementary empirical study (Yi et al., 2023) quantify memorization through mutual information, revealing that empirically optimal models often exhibit poor generalization.

### 2.2 DATA AUGMENTATION FOR GENERATIVE MODELS

The concept of data augmentation in generative modeling was systematically explored by Dist-Aug (Jun et al., 2020), later widely adopted in GAN frameworks during their prominence. Diff-Augment (Zhao et al., 2020) introduced differentiable augmentations specifically optimized for GAN training, while AugSelf-GAN (Hou et al., 2023) enhanced data efficiency through integrated self-supervised tasks. StyleGAN-ADA (Karras et al., 2020) systematically analyzed the effects of augmentation in limited data regimes, with subsequent improvements in the probability of adaptive augmentation via APA (Jiang et al., 2021). While these approaches primarily targeted GANs, recent diffusion models like EDM and EDM2 have successfully adapted conventional augmentation techniques through noise-conditional transformations. Data augmentation naturally allows self-supervised tasks, such as equivariance constraints. In the field of diffusion models, AF-LDM (Zhou et al., 2025) introduces shift-equivariance constraints to mitigate aliasing, while EquiVDM (Liu & Vahdat, 2025) explores temporal equivariance in video diffusion models. However, none of these studies focus on addressing the overfitting issue in diffusion models. From the theoretical aspect, Robbins (2024) analyzes the change of variables for score in different spaces under invertible transformations, while our proposed ScoreAug also adopts and analyzes the noninvertible transformations.

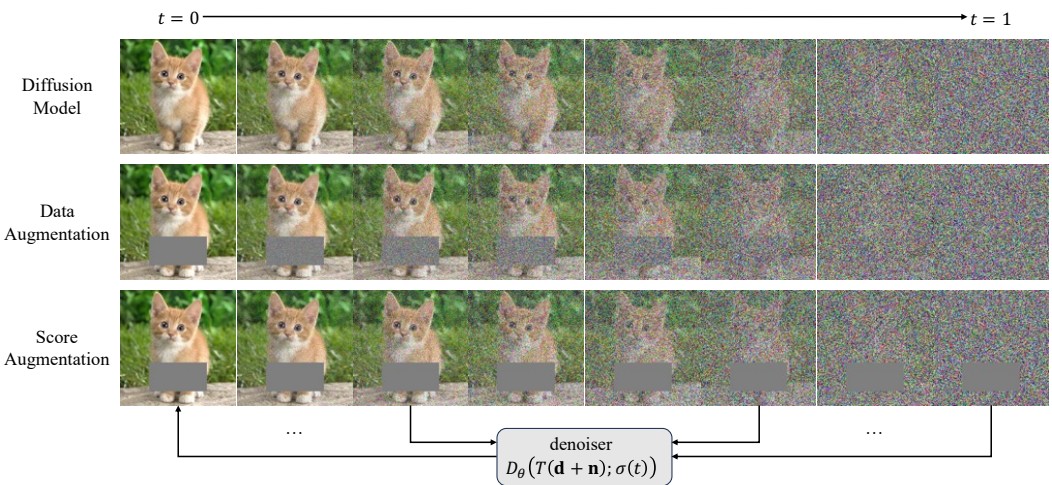

Figure 1: A schematic illustrating the forward process of standard diffusion models, diffusion models with data augmentation, and diffusion models with our proposed Score Augmentation (ScoreAug), as well as the learning objective of ScoreAug.

## 3 PRELIMINARIES

Diffusion models are a family of generative models that consist of a forward noising and a backward denoising process. The forward process of diffusion models typically corresponds to the following forward stochastic differential equation (SDE) (Song et al., 2020b):

$$d\mathbf{x} = f(\mathbf{x}, t)dt + g(t)d\mathbf{w},$$

where the drift coefficient has the form of $f(\mathbf{x}, t) = f(t)\mathbf{x}$ with $f(t) : \mathbb{R} \to \mathbb{R}$, and $g(t) : \mathbb{R} \to \mathbb{R}$ is the diffusion coefficient. Here, $t \in [0, 1]$ is the diffusion time, $\mathbf{x} \in \mathbb{R}^d$ is the data in the $d$-dimensional ambient space, and $\mathbf{w} \in \mathbb{R}^d$ is the standard Wiener process. The perturbation kernels of the SDE have the following form (Karras et al., 2022):

$$p(\mathbf{x}_t \mid \mathbf{x}_0) = \mathcal{N}(\mathbf{x}_t; s(t)\mathbf{x}_0, s(t)^2\sigma(t)^2\mathbf{I}), \tag{1}$$

where $\mathcal{N}(\mathbf{x}; \boldsymbol{\mu}, \boldsymbol{\Sigma})$ is the probability density function of Gaussian distribution with mean $\boldsymbol{\mu} \in \mathbb{R}^d$ and covariance $\boldsymbol{\Sigma} \in \mathbb{R}^{d \times d}$ evaluated at data point $\mathbf{x}$. The coefficients are:

$$s(t) = \exp\left(\int_0^t f(\xi)d\xi\right) \quad \text{and} \quad \sigma(t) = \sqrt{\int_0^t \frac{g(\xi)^2}{s(\xi)^2}d\xi}.$$

According to the different choices of the monotonically decreasing schedule $s(t)$ and the monotonically increasing schedule $\sigma(t)$, diffusion models can be divided into three formulations: 1) variance exploding (VE) $s(t) = 1, \sigma(t) = \sqrt{t}$ such that $s(t)^2 + s(t)^2\sigma(t)^2 > 1, \forall t \in (0, 1]$; 2) variance preserving (VP), $s(t)^2 + s(t)^2\sigma(t)^2 = 1, \forall t \in [0, 1]$; 3) sub-VP, $s(t)^2 + s(t)^2\sigma(t)^2 \leq 1, \forall t \in [0, 1]$. The forward SDE corresponds to the probability flow ordinary differential equation (PF-ODE) (Song et al., 2020b; Karras et al., 2022) that can recover the data distribution $p_{\text{data}}$ from the tractable prior $\mathcal{N}(\mathbf{0}, s(1)^2\sigma(1)^2\mathbf{I})$:

$$d\mathbf{x} = \left[\frac{\dot{s}(t)}{s(t)}\mathbf{x} - s(t)^2\dot{\sigma}(t)\sigma(t)\nabla_{\mathbf{x}} \log p\left(\frac{\mathbf{x}}{s(t)}; \sigma(t)\right)\right]dt, \tag{2}$$

where we have $p(\mathbf{x}; \sigma) \triangleq p_{\text{data}} * \mathcal{N}(\mathbf{0}, \sigma(t)^2\mathbf{I})$ with $*$ the convolution operator, so that the marginal distribution of perturbed data at time $t$ is $p(\mathbf{x}_t) = s(t)^{-d}p(\mathbf{x}_t/s(t); \sigma(t))$. EDM (Karras et al., 2022) leverages a denoiser $D(\cdot; \sigma) : \mathbb{R}^d \to \mathbb{R}^d$ for noise level $\sigma(t)$[1] with the optimization objective:

$$\mathcal{L}_{\text{edm}}(D; \sigma) = \mathbb{E}_{\mathbf{d} \sim p_{\text{data}}, \mathbf{n} \sim \mathcal{N}(\mathbf{0}, \sigma^2\mathbf{I})} \|D(\mathbf{d} + \mathbf{n}; \sigma) - \mathbf{d}\|_2^2, \tag{3}$$

---

[1]We follow EDM to omit $t$ for simplicity when there is no ambiguity in the context.

Table 1: Model input and learning target of the denoiser in different methods.

| Method | Input for denoiser | Target for denoiser |
|---|---|---|
| Diffusion Model | $\mathbf{d} + \mathbf{n}$ | $\mathbf{d}$ |
| Data Augmentation | $T(\mathbf{d}; \boldsymbol{\omega}) + \mathbf{n}$ | $T(\mathbf{d}; \boldsymbol{\omega})$ |
| Score Augmentation (Type I) | $T(\mathbf{d} + \mathbf{n}; \boldsymbol{\omega})$ | $T(\mathbf{d}; \boldsymbol{\omega})$ |
| Score Augmentation (Type II) | $T(\mathbf{d}; \boldsymbol{\omega}) + T(\mathbf{n}; \boldsymbol{\omega})$ | $T(\mathbf{d}; \boldsymbol{\omega})$ |

where the denoiser is typically constructed as $D_{\boldsymbol{\theta}}(\mathbf{x}; \sigma) = c_{\mathrm{skip}}(\sigma)\mathbf{x} + c_{\mathrm{out}}(\sigma)F_{\boldsymbol{\theta}}(c_{\mathrm{in}}(\sigma)\mathbf{x}; c_{\mathrm{noise}}(\sigma))$ with predefined scaling functions $c_{\mathrm{skip}}, c_{\mathrm{out}}, c_{\mathrm{in}}, c_{\mathrm{noise}} : \mathbb{R}_{\geq 0} \to \mathbb{R}$ and a neural network $F_{\boldsymbol{\theta}}(\cdot; \sigma_{\mathrm{noise}}(t)) : \mathbb{R}^d \to \mathbb{R}^d$ with trainable parameters $\boldsymbol{\theta} \in \Theta$. The Stein score (the gradient of log density of the perturbed data distribution $p(\mathbf{x}; \sigma(t))$) can be obtained from the optimal denoiser:

$$\nabla_{\mathbf{x}} \log p(\mathbf{x}; \sigma(t)) = \frac{D_{\boldsymbol{\theta}}(\mathbf{x}; \sigma(t)) - \mathbf{x}}{\sigma(t)^2}, \tag{4}$$

thereby achieving generation of new samples through Eqs. (2) and (4) with the learned denoiser $D_{\boldsymbol{\theta}}$.

## 4 METHOD

### 4.1 DATA AUGMENTATION

The strong memorization capacity of diffusion models manifests as overfitting when the training data is limited, which we observed in experiments (see Fig. 2). As an effective regularization, data augmentation can mitigate overfitting to some extent. In its conventional form, a transformation $T(\cdot; \boldsymbol{\omega}) : \mathbb{R}^d \to \mathbb{R}^d$ (where $\boldsymbol{\omega} \in \boldsymbol{\Omega}$ specifies the augmentation parameters) is applied only to the clean data, i.e., samples at timestep $t = 0$. With augmentation $T$ and noise level $\sigma$, the training objective is:

$$\mathcal{L}_{\mathrm{da}}(D; \sigma, \boldsymbol{\omega}) = \mathbb{E}_{\mathbf{d} \sim p_{\mathrm{data}}, \mathbf{n} \sim \mathcal{N}(\mathbf{0}, \sigma^2 \mathbf{I})} \left\| D\left(T(\mathbf{d}; \boldsymbol{\omega}) + \mathbf{n}; \sigma, \boldsymbol{\omega}\right) - T(\mathbf{d}; \boldsymbol{\omega}) \right\|_2^2. \tag{5}$$

However, it has two drawbacks: (i) augmentation leakage may occur especially with aggressive transforms if the denoiser is not explicitly conditioned on $\boldsymbol{\omega}$ (see Table 5); and (ii) by augmenting only the clean endpoint ($t = 0$), it ignores how the forward noising process acts across timesteps $t > 0$, which can lead to a mismatch with the diffusion dynamics and thus suboptimal regularization.

### 4.2 SCORE AUGMENTATION: TYPE I

We address these issues by performing score augmentation (ScoreAug), in which we treat the transformation $T$ as changing the denoising space instead of the origin data and train the denoiser directly in that transformed space. In other words, we transform the noisy data instead of the clean data, and then input it to the denoiser to let it predict the transformed clean data. Formally, the loss function of ScoreAug is defined as follows:

$$\mathcal{L}_{\mathrm{sa}}^{\mathrm{I}}(D; \sigma, \boldsymbol{\omega}) = \mathbb{E}_{\mathbf{d} \sim p_{\mathrm{data}}, \mathbf{n} \sim \mathcal{N}(\mathbf{0}, \sigma^2 \mathbf{I})} \left\| D\left(T(\mathbf{d} + \mathbf{n}; \boldsymbol{\omega}); \sigma, \boldsymbol{\omega}\right) - T(\mathbf{d}; \boldsymbol{\omega}) \right\|_2^2, \tag{6}$$

and the total objective function is expected for all augmentations and noise levels: $\mathcal{L}(D) = \mathbb{E}_{\boldsymbol{\omega} \sim p_{\Omega}} \mathbb{E}_{\sigma \sim p_{\sigma}} \lambda(\sigma) \mathcal{L}_{\mathrm{sa}}^{\mathrm{I}}(D; \sigma, \boldsymbol{\omega})$, where $p_{\sigma}$ is the prior of noise level (we follow EDM to set $\ln(\sigma) \sim \mathcal{N}(-1.2, 1.2^2)$), $p_{\Omega}$ is the prior of augmentation parameters (we set it to uniform sampling), and $\lambda(\sigma) = (\sigma^2 + \sigma_{\mathrm{data}}^2)/(\sigma \cdot \sigma_{\mathrm{data}})^2$ is the loss weighting that also follows EDM. ScoreAug augments the entire noising trajectory and enforces an equivariant learning target, establishing a harmonious equivariant relationship with the original diffusion formulation while leaving the normal sampling trajectory unaffected, thereby preventing augmentation leakage (see Table 5).

#### 4.2.1 LINEAR TRANSFORMATIONS

Data transformation can generally be divided into linear transformation and nonlinear transformation. We begin with the linear scenario and then give a theoretical analysis of the general case. According

to the perturbation kernel (Eq. (1)) in diffusion models, the forward process can be expressed as $\mathbf{x}_t = s(t)\mathbf{x}_0 + s(t)\sigma(t)\boldsymbol{\epsilon}$, where $\boldsymbol{\epsilon} \sim \mathcal{N}(\mathbf{0}, \mathbf{I})$ is the normal multivariate Gaussian noise. Let augmentation $T$ be a linear transformation that has the form of $T(\mathbf{x}, \boldsymbol{\omega}) = \mathbf{T}_{\boldsymbol{\omega}}\mathbf{x}$ for the corresponding transformation matrix $\mathbf{T}_{\boldsymbol{\omega}} \in \mathbb{R}^{d \times d}$.[2] Each augmentation defines an augmented space, and the forward process in the augmented space is: $\mathbf{y}_t \triangleq T(\mathbf{x}_t; \boldsymbol{\omega}) = s(t)\mathbf{T}\mathbf{x}_0 + s(t)\sigma(t)\mathbf{T}\boldsymbol{\epsilon}$, According to this forward process, we can obtain the perturbation kernel in the augmented space: $p(\mathbf{y}_t \mid \mathbf{y}_0) = \mathcal{N}(\mathbf{y}_t; s(t)\mathbf{y}_0, s(t)^2\sigma(t)^2\mathbf{T}\mathbf{T}^\top)$. For finite data samples $\{\mathbf{x}_1, \ldots, \mathbf{x}_N\}$, where $N \in \mathbb{Z}^+$ is the number of observed training data, the empirical data distribution can be constructed as $p_{\text{data}}(\mathbf{x}) = 1/N \sum_{i=1}^N \delta(\mathbf{x} - \mathbf{x}_i)$ with standard deviation $\sigma_{\text{data}} \in \mathbb{R}^+$, where $\delta(\cdot)$ is the Dirac delta function. Let the corresponding transformed data samples be $\{\mathbf{y}_1, \ldots, \mathbf{y}_N\}$ with the transformed data density $\hat{p}_{\text{data}}(\mathbf{y}) = 1/N \sum_{i=1}^N \delta(\mathbf{y} - \mathbf{y}_i)$ under a given augmentation. And the distribution of the transformed data at time $t$ is $p_t(\mathbf{y}) = \iint_{\mathbb{R}^d \times \mathbb{R}^d} \delta(\mathbf{y} - T(\mathbf{x}_t; \boldsymbol{\omega}))p(\mathbf{x}_t|\mathbf{x}_0)p(\mathbf{x}_0)\mathrm{d}\mathbf{x}_t\mathrm{d}\mathbf{x}_0 = \int_{\mathbb{R}^d} p(\mathbf{y}_t|\mathbf{y}_0)p(\mathbf{y}_0)\mathrm{d}\mathbf{y}_0$. We can then define the distribution of the transformed data at noise level $\sigma$ as: $p(\mathbf{y}; \sigma) = \hat{p}_{\text{data}} * \mathcal{N}(\mathbf{0}, \sigma(t)^2\mathbf{T}\mathbf{T}^\top) = s(t)^d p_{t^{-1}(\sigma)}(s(t)\mathbf{y})$. Under the assumption of infinite model capacity, we can prove (see Section A.1) that the ideal augmented denoiser has the form of:

$$D(\mathbf{y}; \sigma, \boldsymbol{\omega}) = \frac{\sum_{i=1}^Y \mathcal{N}(\mathbf{y}; \mathbf{y}_i, \sigma^2\mathbf{T}\mathbf{T}^\top)\mathbf{y}_i}{\sum_{i=1}^Y \mathcal{N}(\mathbf{y}; \mathbf{y}_i, \sigma^2\mathbf{T}\mathbf{T}^\top)}.$$

The score w.r.t. augmented data $\mathbf{y}$ at noise level $\sigma$ can be obtained from the ideal augmented denoiser:

$$\nabla_\mathbf{y} \log p(\mathbf{y}; \sigma) = (\mathbf{T}\mathbf{T}^\top)^\dagger \frac{D(\mathbf{y}; \sigma, \boldsymbol{\omega}) - \mathbf{y}}{\sigma^2}, \tag{7}$$

where $(\mathbf{T}\mathbf{T}^\top)^\dagger$ is the Moore-Penrose inverse of $\mathbf{T}\mathbf{T}^\top$. If $\mathbf{T}\mathbf{T}^\top$ is a singular matrix, the corresponding Gaussian distribution is a degenerate distribution. The above formula also holds when the gradient $\nabla_\mathbf{y} \log p(\mathbf{y}; \sigma)$ is defined in the image space $\text{Im}(\mathbf{T})$ of the matrix $\mathbf{T}$. From this perspective, it becomes evident that ScoreAug essentially requires the optimal denoiser to be equivariant with respect to the employed linear transformation, as written as:

$$D(\mathbf{T}\mathbf{x}; \sigma) = \mathbf{T}\mathbf{x} + \sigma^2\mathbf{T}\mathbf{T}^\top\nabla_{\mathbf{T}\mathbf{x}} \log p(\mathbf{T}\mathbf{x}; \sigma) = \mathbf{T}D(\mathbf{x}; \sigma).$$

For $\mathbf{y} = T(\mathbf{x})$, where $T$ is linear and invertible, we have $\nabla_\mathbf{y} \log p(\mathbf{y}; \sigma) = \mathbf{T}^{-\top}\nabla_\mathbf{x} \log p(\mathbf{x}; \sigma)$ that reveals the correspondence between scores in transformed spaces (see Theorem 1 for general case). When combined with Eqs. (4) and (7), this demonstrates that the new denoiser learns scores in different spaces, essentially different from data augmentation.

**Augmentation and Condition** We borrow the practices of (Zhao et al., 2020; Hou et al., 2023; Karras et al., 2020) to adopt linear transformations (brightness, translation, cutout, and rotation) as data transformations. We empirically find that any independent augmentation can improve the performance of the baseline, and the combination is significantly better (see Table 4).

Let $H \in \mathbb{Z}^+$ and $W \in \mathbb{Z}^+$ be the height and width of the image, respectively. The transformation matrix corresponding to different augmentations. **Brightness** scales the images by $\omega_b \in [1/B, B]$ with $B \in \mathbb{R}^+$, such that $T_{ij}^{\omega_b} = \{\omega_b \text{ if } i = j, \ 0 \text{ otherwise}\}$. **Translation** shifts images by $\Delta_i \in \{1, \cdots, \lfloor R_t W \rfloor\}$ vertical and $\Delta_j \in \{1, \cdots, \lfloor R_t H \rfloor\}$ horizontal pixels with $R_t \in \mathbb{R}^+$, where $\boldsymbol{\omega}_t = (\Delta_i, \Delta_j)$, such that $T_{ij}^{\boldsymbol{\omega}_t} = \{1 \text{ if } i = j + \Delta i + H \cdot \Delta j, \ 0 \text{ otherwise}\}$. **Cutout** zeros a rectangular region centered at point of $(c_x, c_y)$ with size of $(h, w)$ that $h \in \{1, \cdots, \lfloor R_c H \rfloor\}, w \in \{1, \cdots, \lfloor R_c W \rfloor\}$ with $R_c \in \mathbb{R}^+$, where $\boldsymbol{\omega}_c = (c_x, c_y, h, w)$, such that $T_{ij}^{\boldsymbol{\omega}_c} = \{1 \text{ if } \left|\frac{i}{W} - c_y\right| > \frac{h}{2} \text{ or } |(i \bmod W) - c_x| > \frac{w}{2}, \ 0 \text{ otherwise}\}$. **Rotation** rotates images by $90° \times \omega_r$, where $\omega_r \in \{0, 1, 2, 3\}$, such that $T_{ij}^{\omega_r} = \{1 \text{ if } i = (H - 1 - h) + wW, j = h + wH, \ 0 \text{ otherwise}\}$.

Note that translation and cropping are zero-padded instead of masked. The difference is that we calculate the loss of the padded area, while the mask does not. The augmentation parameters are randomly sampled from predetermined ranges that include identity mapping to ensure learning from the original data. For the condition input (if any) to the denoiser, ScoreAug add a linear layer to directly accept the condition vector $\boldsymbol{\omega}$, and then add it together with the timestep embedding. For cutout, the center point coordinate $(c_x, c_y)$ is removed and only the cutout size $\boldsymbol{\omega}_c = (h, w)$ is kept. When sampling, we can set the condition (if any) to an appropriate value (e.g., zeros) to generate an untransformed image and prevent augmentation-leaking for aggressive augmentations (see Table 5).

---

[2]For notational simplicity, we may omit $\boldsymbol{\omega}$ in $T(\cdot; \boldsymbol{\omega})$ and $\mathbf{T}_{\boldsymbol{\omega}}$ when the context is unambiguous.

Table 2: FID comparisons between ScoreAug and EDM on unconditional and conditional CIFAR-10, FFHQ, and AFHQv2. NLA means non-leaky augmentation on original data. VP and VE means variance-preserving and variance-exploding, respectively. NFE means the number of function evaluations. We reproduce the results of EDM using the official code for a fair comparison.

| Method | CIFAR-10 $32 \times 32$ | | | | FFHQ $64 \times 64$ | | AFHQv2 $64 \times 64$ | |
| | Unconditional | | Conditional | | Unconditional | | Unconditional | |
| | VP | VE | VP | VE | VP | VE | VP | VE |
|---|---|---|---|---|---|---|---|---|
| EDM w/o NLA | 4.05 | 4.10 | 4.03 | 4.32 | 5.26 | 4.98 | 5.69 | 5.58 |
| + dropout$\times 2$ | 3.13 | 2.93 | 2.93 | 2.77 | 4.87 | 4.63 | 4.60 | 4.54 |
| + weight decay | 3.13 | 3.01 | 3.17 | 2.93 | 4.76 | 4.69 | 5.76 | 4.93 |
| + ScoreAug (Linear) | **2.35** | **2.24** | **2.11** | **2.25** | **2.96** | **2.88** | **3.55** | **3.54** |
| EDM w/ NLA | 2.07 | 2.10 | 1.93 | 1.92 | 2.76 | 2.80 | 2.65 | 2.68 |
| + ScoreAug (Linear) | **2.05** | 2.06 | **1.80** | 1.91 | 2.72 | **2.69** | **2.30** | **2.18** |
| $\times$ ScoreAug (Type I) | **2.05** | **1.96** | 1.90 | **1.81** | **2.63** | 2.89 | 2.70 | 2.68 |
| $\times$ ScoreAug (Type II) | 2.06 | 1.97 | 1.85 | 1.96 | 2.76 | 3.02 | 2.37 | 2.58 |
| NFE | 35 | 35 | 35 | 35 | 79 | 79 | 79 | 79 |

### 4.2.2 THEORETICAL ANALYSIS

**Theorem 1 (Transformation of Score Functions)** *Let $p(\mathbf{x})$ be the probability density function (PDF) of $\mathbf{x} \in \mathbb{R}^n$, $\mathbf{y} = T(\mathbf{x})$ for a differentiable map $T : \mathbb{R}^n \to \mathbb{R}^m$ ($m \leq n$) with Jacobian $\mathbf{J}_T(\mathbf{x})$ and full row rank $m$ on the relevant support and let $p(\mathbf{y})$ be the PDF of $\mathbf{y}$. Then we have:*

$$\nabla_{\mathbf{y}} \log p(\mathbf{y}) = \mathbb{E}_{p(\mathbf{x}|\mathbf{y})}\left[\mathbf{J}_T(\mathbf{x})^{\dagger}\left(\nabla_{\mathbf{x}} \log p(\mathbf{x}) - \tfrac{1}{2}\nabla_{\mathbf{x}} \log \det\left(\mathbf{J}_T(\mathbf{x})\mathbf{J}_T(\mathbf{x})^{\top}\right)\right)\right].$$

***Diffeomorphism***: *If $m = n$ and $T$ is a (global) diffeomorphism with $\mathbf{x} = T^{-1}(\mathbf{y})$:*

$$\nabla_{\mathbf{y}} \log p(\mathbf{y}) = \mathbf{J}_T(\mathbf{x})^{-\top}(\nabla_{\mathbf{x}} \log p(\mathbf{x}) - \nabla_{\mathbf{x}} \log |\det \mathbf{J}_T(\mathbf{x})|).$$

***Linear Surjection***: *If $T(\mathbf{x}) = \mathbf{T}\mathbf{x}$, where $\mathbf{T} \in \mathbb{R}^{m \times n}$ is a constant matrix with $\mathrm{rank}(\mathbf{T}) = m$:*

$$\nabla_{\mathbf{y}} \log p(\mathbf{y}) = (\mathbf{T}\mathbf{T}^{\top})^{-1}\mathbf{T} \cdot \mathbb{E}_{p(\mathbf{x}|\mathbf{y})}[\nabla_{\mathbf{x}} \log p(\mathbf{x})].$$

Theorem 1 (proved in Section A.2) first establishes the correspondence between score functions in different spaces under general transformations and gives two special case versions under diffeomorphism and linear surjection, respectively. Since the denoiser learns the score function in a given space, ScoreAug can be regarded as learning score functions in different spaces. This imposes an implicit regularization on the denoiser, thereby improving its generalization ability to alleviate overfitting.

### 4.3 SCORE AUGMENTATION: TYPE II

When the transformation $T$ is linear, by the forward formula of diffusion models, applying $T$ to $\mathbf{x}_t$ is equivalent to applying $T$ to $\mathbf{x}_0$ and to $\boldsymbol{\epsilon}$ separately and then adding the results. This motivates the second type of loss function for ScoreAug, given by

$$\mathcal{L}_{\mathrm{sa}}^{\mathrm{II}}(D; \sigma, \boldsymbol{\omega}) = \mathbb{E}_{\mathbf{d} \sim p_{\mathrm{data}}, \mathbf{n} \sim \mathcal{N}(\mathbf{0}, \sigma^2 \mathbf{I})} \|D(T(\mathbf{d}; \boldsymbol{\omega}) + T(\mathbf{n}; \boldsymbol{\omega}); \sigma) - T(\mathbf{d}; \boldsymbol{\omega})\|_2^2. \tag{8}$$

We refer to Eq. (6) as ScoreAug (Type I) and to Eq. (8) as ScoreAug (Type II). When $T$ is linear, the two are equivalent; when $T$ is nonlinear, they differ: a first-order Taylor expansion gives $T(\mathbf{d} + \mathbf{n}; \boldsymbol{\omega}) \approx T(\mathbf{d}; \boldsymbol{\omega}) + \mathbf{J}_T(\mathbf{d})\mathbf{n}$, which indicates that Type II can be interpreted as an additive-noise surrogate of Type I and, moreover, avoids state-dependent heteroscedasticity. We evaluate both variants under nonlinear data transformations and report their comparative performance in Table 2. Empirically, each shows advantages in different settings, and we leave a deeper investigation of their fundamental differences to future work.

Table 3: Quantitative comparisons between SiT and ScoreAug on ImageNet-256.

| Dataset | Model | Steps | FID ↓ | sFID ↓ | IS ↑ | Precision ↑ | Recall ↑ |
|---------|-------|-------|-------|--------|------|-------------|----------|
| ImageNet-256 | SiT-XL | 400K | 19.26 | 5.24 | 70.75 | 0.6223 | **0.6436** |
| | + ScoreAug | 400K | **18.75** | **5.21** | **71.79** | **0.6249** | 0.6372 |
| | SiT-XL | 1M | 13.21 | 5.39 | 94.64 | 0.6542 | **0.6599** |
| | + ScoreAug | 1M | **12.70** | **5.36** | **96.37** | **0.6589** | 0.6581 |

Table 4: Ablation study of ScoreAug under individual and combined augmentations. FID scores are reported for both unconditional and conditional CIFAR-10 settings across the variance preserving and variance exploding paradigms.

| Method | Augmentation | | | | Uncond CIFAR | | Cond CIFAR | |
|--------|------------|-------------|--------|----------|------|------|------|------|
| | brightness | translation | cutout | rotation | VP | VE | VP | VE |
| EDM | - | - | - | - | 4.05 | 4.10 | 4.03 | 4.32 |
| + ScoreAug | ✓ | ✗ | ✗ | ✗ | 2.97 | 2.85 | 2.86 | 2.68 |
| + ScoreAug | ✗ | ✓ | ✗ | ✗ | 2.68 | 2.86 | 2.40 | 2.62 |
| + ScoreAug | ✗ | ✗ | ✓ | ✗ | 3.68 | 3.56 | 3.62 | 3.24 |
| + ScoreAug | ✗ | ✗ | ✗ | ✓ | 2.43 | 2.69 | 2.13 | 2.59 |
| + ScoreAug | ✓ | ✓ | ✓ | ✓ | **2.27** | **2.29** | **2.11** | **2.06** |

## 5 EXPERIMENTS

### 5.1 EXPERIMENTAL SETUP

We implement the proposed ScoreAug based on the official EDM code[3] due to its generality. We follow the settings of EDM, including the network and preconditioning, training, sampling, and parameters (see Table 1 in the EDM paper (Karras et al., 2022)), to conduct experiments for a fair comparison with the baselines. The datasets are unconditional CIFAR-10 and conditional CIFAR-10 (Krizhevsky et al., 2009), FFHQ (Karras et al., 2019), and AFHQv2 (Choi et al., 2020). On each dataset, we experiment with both variance preserving (VP) (Nichol & Dhariwal, 2021; Karras et al., 2022) and variance exploding (VE) (Song & Ermon, 2019; Karras et al., 2022) formulations. Fréchet Inception Distance (FID) (Heusel et al., 2017) is used as the main evaluation metric. All models are trained for 200,000 iterations with batch size of 512. All results are calculated from the model evaluation at the last scheduled checkpoint for fair comparisons with 50,000 generated images unless otherwise specified. In the experiments, we use the code officially provided by EDM and re-run its results as a baseline for an absolutely fair comparison.

### 5.2 MAIN RESULTS

Table 2 presents comparative results between ScoreAug and competing methods. The baseline, denoted as EDM w/o NLA (without non-leaking augmentation), exhibits clear signs of overfitting, as evidenced by performance gains when increasing dropout or incorporating weight decay. In contrast, integrating ScoreAug significantly outperforms these basic overfitting mitigation strategies, underscoring the efficacy of our approach. Notably, EDM w/ NLA, which employs sophisticated non-linear augmentations, achieves superior results. Remarkably, ScoreAug's non-linear extension can seamlessly generalize to these augmentations. Both variants (Type I Eq. (6) and Type II Eq. (8)) attain improved FID scores across most scenarios, demonstrating their expansibility. Furthermore, the linear variant of ScoreAug can be applied synergistically to EDM w/ NLA, yielding additional performance improvements. This flexibility highlights the broad applicability of our method, even when integrated with state-of-the-art augmentation frameworks. Section E show the images generated by EDM and ScoreAug trained on AFHQ-v2 and FFHQ, respectively. In the following, we default to using ScoreAug with linear augmentation for experiments unless otherwise specified.

---

[3]https://github.com/NVlabs/edm

Table 5: FID scores of ScoreAug without or with conditioning under different augmentation combination combinations on unconditional and conditional CIFAR-10 across VP and VE settings.

| Method | Augmentation | | | Uncond CIFAR | | Cond CIFAR | |
|---|---|---|---|---|---|---|---|
| | translation | cutout | rotation | VP | VE | VP | VE |
| EDM | - | - | - | 4.05 | 4.10 | 4.03 | 4.32 |
| + DataAug w/o condition | ✓ | ✓ | ✗ | 11.02 | 10.87 | 10.31 | 10.36 |
| + ScoreAug w/o condition | ✓ | ✓ | ✗ | 2.53 | 2.63 | 2.41 | 2.37 |
| + ScoreAug w/ condition | ✓ | ✓ | ✗ | 2.48 | 2.55 | 2.41 | 2.55 |
| + ScoreAug w/o condition | ✓ | ✓ | ✓ | 22.90 | 25.15 | 24.29 | 23.68 |
| + ScoreAug w/ condition | ✓ | ✓ | ✓ | **2.21** | **2.12** | **2.01** | **2.08** |

**Diffusion Transformer Architecture** We also conducted experiments on ImageNet-256 (Russakovsky et al., 2015) using the diffusion transformer architecture (Peebles & Xie, 2023), adopting the state-of-the-art SiT (Ma et al., 2024) training code[4] with an XL-scale model configuration. Evaluation metrics include FID (Heusel et al., 2017), sFID (Nash et al., 2021), IS (Salimans et al., 2016), as well as precision and recall (Sajjadi et al., 2018). As evidenced in Table 3, ScoreAug demonstrates superior performance over SiT-XL in most metrics at 400K and 1M training steps, substantiating its effectiveness when applied to advanced model architectures and large complex datasets.

### 5.3 ANALYSIS EXPERIMENTS

**Different Augmentations** Note that ScoreAug is compatible with multiple linear data augmentation types by randomly selecting one augmentation per training iteration. We investigated the impact of individual augmentations (brightness, translation, cutout, rotation) and their combinations on model performance. As shown in Table 4, all single augmentations outperform the no-augmentation baseline, and combined usage achieves the best results, demonstrating synergistic effects. Based on the experimental results, we posit that incorporating more linear data augmentations can further improve performance, and we leave this exploration for future work.

**Importance of Conditioning** In the section introducing ScoreAug, we default to incorporating the augmentation parameter $\omega$ as the condition for the model to distinguish between data augmentation types and intensities. However, this conditioning is not strictly necessary. As reported in Table 5, ScoreAug remains functional without conditioning if the augmentations are non-invertible, such as translation and cutout. However, data augmentation without conditioning (DataAug w/o condition) causes data leakage in this case, leading to a significant drop in FID. Conversely, for invertible augmentations like uniform random rotation by $\{0°, 90°, 180°, 270°\}$ (Karras et al., 2020; Hou et al., 2021), unconditional ScoreAug fails to generate rotated images, resulting in augmentation-leaking issue. This failure arises because the random noise distribution is rotation-invariant, causing ScoreAug to treat rotated images as original training data, effectively reducing it to standard data augmentation.

Based on these findings, we recommend adding conditions to support broader augmentation types, though this modifies the network architecture. For users aiming to finetune pre-trained models, unconditional ScoreAug remains viable if non-invertible augmentations are adopted. Additionally, conditional injection enables augmentation-controllable generation. For example, synthesizing images with specified rotation angles, as illustrated in Figs. 3 and 4.

**Training Data** Overfitting typically stems from insufficient data and low data utilization efficiency of models. To validate this, we reduced the CIFAR-10 training data to $N = 10,000$ and $N = 20,000$ samples, comparing ScoreAug against the baseline. Results in Figs. 2a and 2d show that baseline performance degrades sharply with smaller datasets, while ScoreAug consistently outperforms it, demonstrating stronger data utilization efficiency and robustness against overfitting.

**Model Size** Another factor influencing model overfitting is model complexity – larger models generally tend to overfit more easily. We compared ScoreAug and the baseline across varying model

---

[4] https://github.com/willisma/SiT

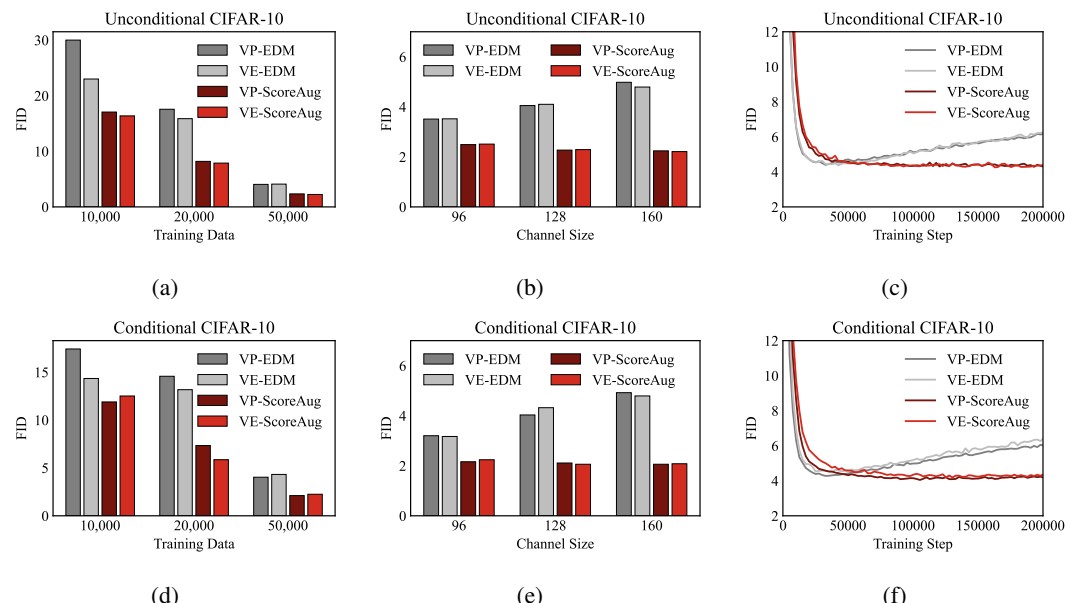

Figure 2: FID comparisons of ScoreAug with EDM in VP and VE diffusion formulations on unconditional and conditional CIFAR-10 datasets: (a,d) different training data sizes; (b,e) different model sizes (channels); (c,f) different training steps (FID is evaluated with 10,000 samples).

sizes by adjusting the number of base channels $C \in \mathbb{N}^+$, which is defaultly set as $C = 128$ in EDM. We tested two additional configurations ($C = 96$ and $C = 160$) to decrease or increase the model size. As shown in Figs. 2b and 2e, the baseline EDM performance degrades with increasing model size, further indicating overfitting issues, while our ScoreAug improves significantly, highlighting its robustness against overfitting.

**Training Convergence**   To visually demonstrate the overfitting issue in diffusion models, we evaluated FID scores at each training checkpoint. In order to reduce the evaluation computational overhead, the FID scores are reduced to 10,000 generated images for calculation, which just matches the number of test images of CIFAR-10. As illustrated in Figs. 2c and 2f, the FID score of EDM initially decreases rapidly to a minimum and then gradually rises during training, indicating overfitting issues. In contrast, our ScoreAug maintains a consistent downward trend in FID scores, effectively mitigating overfitting throughout the training process. Although early stopping allows the baseline to achieve decent FID scores, it still underperforms ScoreAug at their best. Furthermore, early stopping necessitates continuous model evaluation, resulting in additional computational overhead. Overall, our method achieves more stable convergence properties.

## 6   CONCLUSION

In summary, this work reveals and substantiates the risk of overfitting in diffusion models, particularly under data-constrained conditions. To mitigate this issue, we introduced Score Augmentation (Score-Aug), a novel diffusion-aligned augmentation framework that operates directly in the noisy data space. By enforcing an equivariant objective under data transformations, ScoreAug integrates naturally with the denoising process and extends gracefully to both linear and non-linear transformations. Extensive experiments across CIFAR-10, FFHQ, AFHQv2, and ImageNet, with U-Net and DiT backbones, confirm its effectiveness in enhancing generation quality, improving robustness against overfitting, and maintaining stable training dynamics. Moreover, ScoreAug is complementary to conventional augmentation techniques, enabling additional gains when combined. We believe these findings highlight the importance of data-space design in diffusion training and open up promising directions for leveraging equivariant learning in generative modeling.

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

## A  DERIVATION AND PROOF

### A.1  DIFFUSION MODELS IN LINEAR AUGMENTED SPACE

**Optimal Denoiser**   For the loss function in Eq. (6), its expansion yields:

$$\mathcal{L}(D;\sigma,\boldsymbol{\omega}) = \mathbb{E}_{\mathbf{d}\sim p_{\text{data}}}\mathbb{E}_{\mathbf{n}\sim\mathcal{N}(\mathbf{0},\sigma^2\mathbf{I})}\|D(T(\mathbf{d}+\mathbf{n};\boldsymbol{\omega});\sigma,\boldsymbol{\omega}) - T(\mathbf{d};\boldsymbol{\omega})\|_2^2$$
$$= \mathbb{E}_{\mathbf{d}\sim p_{\text{data}}}\mathbb{E}_{\mathbf{x}\sim\mathcal{N}(\mathbf{d},\sigma^2\mathbf{I})}\|D(T(\mathbf{x};\boldsymbol{\omega});\sigma,\boldsymbol{\omega}) - T(\mathbf{d};\boldsymbol{\omega})\|_2^2$$
$$= \mathbb{E}_{\mathbf{d}\sim p_{\text{data}}}\mathbb{E}_{\mathbf{x}\sim\mathcal{N}(\mathbf{d},\sigma^2\mathbf{I})}\|D(\mathbf{Tx};\sigma,\boldsymbol{\omega}) - \mathbf{Td}\|_2^2$$
$$= \mathbb{E}_{\mathbf{y}_0\sim\hat{p}_{\text{data}}}\mathbb{E}_{\mathbf{y}\sim\mathcal{N}(\mathbf{y}_0,\sigma^2\mathbf{TT}^\top)}\|D(\mathbf{y};\sigma,\boldsymbol{\omega}) - \mathbf{y}_0\|_2^2$$
$$= \int_{\mathbb{R}^d}\frac{1}{N}\sum_{i=1}^N\mathcal{N}(\mathbf{y};\mathbf{y}_i,\sigma^2\mathbf{TT}^\top)\|D(\mathbf{y};\sigma,\boldsymbol{\omega}) - \mathbf{y}_i\|_2^2\mathrm{d}\mathbf{y}$$

To obtain the optimal denoiser, we minimize it independently for each $\mathbf{y}$. Being a convex optimization problem, we set its derivative to zero and obtain the following.

$$0 = \nabla_{D(\mathbf{y};\sigma,\boldsymbol{\omega})}\left[\mathcal{L}(D;\mathbf{y},\sigma,\boldsymbol{\omega})\right]$$
$$0 = \nabla_{D(\mathbf{y};\sigma,\boldsymbol{\omega})}\left[\frac{1}{Y}\sum_{i=1}^Y\mathcal{N}(\mathbf{y};\mathbf{y}_i,\sigma^2\mathbf{TT}^\top)\|D(\mathbf{y};\sigma,\boldsymbol{\omega}) - \mathbf{y}_i\|_2^2\right]$$
$$0 = \sum_{i=1}^Y\mathcal{N}(\mathbf{y};\mathbf{y}_i,\sigma^2\mathbf{TT}^\top)\nabla_{D(\mathbf{y};\sigma,\boldsymbol{\omega})}\left[\|D(\mathbf{y};\sigma,\boldsymbol{\omega}) - \mathbf{y}_i\|_2^2\right]$$
$$0 = \sum_{i=1}^Y\mathcal{N}(\mathbf{y};\mathbf{y}_i,\sigma^2\mathbf{TT}^\top)\left[D(\mathbf{y};\sigma,\boldsymbol{\omega}) - \mathbf{y}_i\right]$$
$$D(\mathbf{y};\sigma,\boldsymbol{\omega}) = \frac{\sum_{i=1}^Y\mathcal{N}(\mathbf{y};\mathbf{y}_i,\sigma^2\mathbf{TT}^\top)\mathbf{y}_i}{\sum_{i=1}^Y\mathcal{N}(\mathbf{y};\mathbf{y}_i,\sigma^2\mathbf{TT}^\top)}$$

**Score Function**   For the transformation $\mathbf{y} = \mathbf{Tx}$, its score function can be expressed as:

$$\nabla_\mathbf{y}\log p(\mathbf{y};\sigma) = \frac{\nabla_\mathbf{y}p(\mathbf{y};\sigma)}{p(\mathbf{y};\sigma)} = \frac{\sum_{i=1}^Y\nabla_\mathbf{y}\mathcal{N}(\mathbf{y};\mathbf{y}_i,\sigma^2\mathbf{TT}^\top)}{\sum_{i=1}^Y\mathcal{N}(\mathbf{y};\mathbf{y}_i,\sigma^2\mathbf{TT}^\top)}$$
$$= (\mathbf{TT}^\top)^\dagger\left(\frac{\sum_{i=1}^Y\mathcal{N}(\mathbf{y};\mathbf{y}_i,\sigma^2\mathbf{TT}^\top)\mathbf{y}_i}{\sum_{i=1}^Y\mathcal{N}(\mathbf{y};\mathbf{y}_i,\sigma^2\mathbf{TT}^\top)} - \mathbf{y}\right)/\sigma^2 \qquad (9)$$
$$= (\mathbf{TT}^\top)^\dagger\left(D(\mathbf{y};\sigma,\boldsymbol{\omega}) - \mathbf{y}\right)/\sigma^2$$

where Eq. (9) comes from $\nabla_\mathbf{y}\mathcal{N}(\mathbf{y};\mathbf{y}_i,\sigma^2\mathbf{TT}^\top) = \mathcal{N}(\mathbf{y};\mathbf{y}_i,\sigma^2\mathbf{TT}^\top)(\mathbf{TT}^\top)^\dagger(\mathbf{y}_i - \mathbf{y})/\sigma^2$.

### A.2  TRANSFORMATION OF SCORE FUNCTIONS UNDER SURJECTIVE SMOOTH MAPPINGS

We have a random vector $\mathbf{x}\in\mathbb{R}^n$ with smooth, positive density $p(\mathbf{x};\sigma)$, and a smooth surjective map

$$T:\mathbb{R}^n\to\mathbb{R}^m, m\le n,$$

of full row-rank $m$ on the support of interest. Write $\mathbf{y} = T(\mathbf{x})$ and denote by

$$p(\mathbf{y};\sigma) = \int p(\mathbf{x};\sigma)\delta(\mathbf{y} - T(\mathbf{x}))\mathrm{d}\mathbf{x}.$$

its marginal density. Since everything is smooth and ppp decays at infinity, we may exchange $\nabla_\mathbf{y}$ and the integral:

$$\nabla_\mathbf{y}p(\mathbf{y};\sigma) = \int p(\mathbf{x};\sigma)\nabla_\mathbf{y}\delta(\mathbf{y} - T(\mathbf{x}))\mathrm{d}\mathbf{x}.$$

Next, use the chain-rule for the delta:

$$\nabla_\mathbf{x}\delta(\mathbf{y} - T(\mathbf{x})) = -\mathbf{J}_T(\mathbf{x})^\top\nabla_\mathbf{y}\delta(\mathbf{y} - T(\mathbf{x})),$$

where $\mathbf{J}_T(\mathbf{x})$ is the $m \times n$ Jacobian matrix of $T$ at $\mathbf{x}$. Since $\mathbf{J}_T(\mathbf{x})$ has full row rank $m$, the $m \times m$ matrix $\mathbf{J}_T(\mathbf{x})\mathbf{J}_T(\mathbf{x})^\top$ is invertible. We can solve for $\delta(\mathbf{y} - T(\mathbf{x}))$:

$$-(\mathbf{J}_T(\mathbf{x})\mathbf{J}_T(\mathbf{x})^\top)^{-1}\mathbf{J}_T(\mathbf{x})\nabla_\mathbf{x}\delta(\mathbf{y} - T(\mathbf{x})) = \nabla_\mathbf{y}\delta(\mathbf{y} - T(\mathbf{x})).$$

Let $\mathcal{J} = (\mathbf{J}_T(\mathbf{x})\mathbf{J}_T(\mathbf{x})^\top)^{-1}\mathbf{J}_T(\mathbf{x})$. Note that $\mathcal{J}$ is the transpose of Moore-Penrose pseudo-inverse $(\mathbf{J}_T(\mathbf{x})^\dagger)^\top$. Substitute back into the expression for $\nabla_\mathbf{y}p(\mathbf{y}; \sigma)$:

$$\nabla_\mathbf{y}p(\mathbf{y}; \sigma) = -\int p(\mathbf{x}; \sigma)\mathcal{J}\nabla_\mathbf{x}\delta(\mathbf{y} - T(\mathbf{x}))\mathrm{d}\mathbf{x}.$$

Let's look at the $i$-th component of $\nabla_\mathbf{y}p(\mathbf{y}; \sigma)$:

$$(\nabla_\mathbf{y}p(\mathbf{y}; \sigma))_i = -\int p(\mathbf{x}; \sigma)\sum_k \mathcal{J}_{ik}\frac{\partial\delta(\mathbf{y} - T(\mathbf{x}))}{\partial x_k}\mathrm{d}\mathbf{x}.$$

Using integration by parts:

$$(\nabla_\mathbf{y}p(\mathbf{y}; \sigma))_i = -\sum_k \int \frac{\partial(p(\mathbf{x}; \sigma)\mathcal{J}_{ik})}{\partial x_k}\delta(\mathbf{y} - T(\mathbf{x}))\mathrm{d}\mathbf{x}$$

$$= -\sum_k \int \left(\frac{\partial p(\mathbf{x}; \sigma)}{\partial x_k}\mathcal{J}_{ik}\right)\delta(\mathbf{y} - T(\mathbf{x}))\mathrm{d}\mathbf{x}.$$

Let $\mathrm{Div}_\mathrm{rows}(M)_i = \sum_k \frac{\partial M_{ik}}{\partial x_k}$. This term represents the divergence of each row of $M$.

$$\nabla_\mathbf{y}\log p(\mathbf{y}; \sigma) = \frac{\nabla_\mathbf{y}p(\mathbf{y}; \sigma)}{p(\mathbf{y}; \sigma)} = \mathbb{E}_{p(\mathbf{x}|\mathbf{y};\sigma)}\left[\mathcal{J}\frac{\nabla_\mathbf{y}p(\mathbf{x}; \sigma)}{p(\mathbf{x}; \sigma)} + \mathrm{Div}_\mathrm{rows}(\mathcal{J})\right].$$

The matrix calculus identities are known (proven later):

$$\mathrm{Div}_\mathrm{rows}(\mathcal{J}) = -\mathcal{J}\frac{1}{2}\nabla_\mathbf{x}\log\det(\mathbf{J}_T(\mathbf{x})\mathbf{J}_T(\mathbf{x})^\top).$$

Substituting this into the expression:

$$\nabla_\mathbf{y}\log p(\mathbf{y}; \sigma) = \mathbb{E}_{p(\mathbf{x}|\mathbf{y};\sigma)}\left[\mathcal{J}\left(\nabla_\mathbf{x}\log p(\mathbf{x}; \sigma) - \frac{1}{2}\nabla_\mathbf{x}\log\det(\mathbf{J}_T(\mathbf{x})\mathbf{J}_T(\mathbf{x})^\top)\right)\right].$$

If $T$ is a global diffeomorphism then $\mathbf{J}_T(\mathbf{x})$ is square and invertible, $\mathcal{J} = \mathbf{J}_T(\mathbf{x})^{-\top}$, and

$$\frac{1}{2}\nabla_\mathbf{x}\log\det(\mathbf{J}_T(\mathbf{x})\mathbf{J}_T(\mathbf{x})^\top) = \nabla_\mathbf{x}\log|\det\mathbf{J}_T(\mathbf{x})|.$$

Hence,

$$\nabla_\mathbf{y}\log p(\mathbf{y}; \sigma) = \mathbf{J}_T(\mathbf{x})^{-\top}\left(\nabla_\mathbf{x}\log p(\mathbf{x}; \sigma) - \nabla_\mathbf{x}\log|\det(\mathbf{J}_T(\mathbf{x}))|\right).$$

If $T(\mathbf{x}) = \mathbf{T}\mathbf{x}$, where $\mathbf{T}$ is a constant $m \times n$ matrix with full row rank $m$:

$$\mathcal{J} = (\mathbf{T}\mathbf{T}^\top)^{-1}\mathbf{T}, \frac{1}{2}\nabla_\mathbf{x}\log\det(\mathbf{J}_T(\mathbf{x})\mathbf{J}_T(\mathbf{x})^\top) = 0.$$

Hence,

$$\nabla_\mathbf{y}\log p(\mathbf{y}; \sigma) = (\mathbf{T}\mathbf{T}^\top)^{-1}\mathbf{T} \cdot \mathbb{E}_{p(\mathbf{x}|\mathbf{y};\sigma)}\left[\nabla_\mathbf{x}\log p(\mathbf{x}; \sigma)\right].$$

**The matrix calculus identities** Since the Jacobian determinant is defined on a smooth function, satisfying $\partial_i\mathbf{J}_{kj} = \partial_j\mathbf{J}_{ki}$, we define the right inverse $\mathbf{J}^\dagger = \mathbf{J}^\top(\mathbf{J}\mathbf{J}^\top)^{-1}$. Prove that:

$$\mathrm{Div}_\mathrm{rows}((\mathbf{J}^\dagger)^\top) = -\mathbf{J}^{\dagger\top}\frac{1}{2}\nabla_\mathbf{x}\log\det(\mathbf{J}\mathbf{J}^\top).$$

This is equivalent to proving:

$$\mathbf{J}^\top\mathrm{Div}_\mathrm{rows}((\mathbf{J}^\dagger)^\top) = -\frac{1}{2}\nabla_\mathbf{x}\log\det(\mathbf{J}\mathbf{J}^\top).$$

Consider the $k$-th term on the left:

$$\text{LHS}_k = \sum_i \mathbf{J}_{ik} \left( \sum_j \frac{\partial ((\mathbf{J}^\dagger)^\top)_{ij}}{\partial x_j} \right) = \sum_{ij} \mathbf{J}_{ik} \partial_j \mathbf{J}_{ji}^\dagger.$$

The $k$-th term on the right side:

$$\begin{aligned}
\text{RHS}_k &= -\frac{1}{2} \frac{\partial \ln \det(\mathbf{J}\mathbf{J}^\top)}{\partial x_k} \\
&= -\frac{1}{2} \sum_{ij} \left( \frac{\partial \ln \det(\mathbf{J}\mathbf{J}^\top)}{\partial \mathbf{J}} \right)_{ij} \frac{\partial \mathbf{J}_{ij}}{\partial x_k} \\
&= -\sum_{ij} (\mathbf{J}^{\dagger\top})_{ij} \frac{\partial \mathbf{J}_{ij}}{\partial x_k} = -\sum_{ij} \mathbf{J}_{ji}^\dagger \partial_j \mathbf{J}_{ik}.
\end{aligned}$$

Hence,

$$\begin{aligned}
\text{LHS}_k - \text{RHS}_k &= \sum_{ij} \mathbf{J}_{ik} \partial_j \mathbf{J}_{ji}^\dagger + \mathbf{J}_{ji}^\dagger \partial_j \mathbf{J}_{ik} \\
&= \sum_{ij} \partial_j (\mathbf{J}_{ji}^\dagger \mathbf{J}_{ik}) \\
&= \sum_j \partial_j \left( \sum_i \mathbf{J}_{ji}^\dagger \mathbf{J}_{ik} \right) = 0,
\end{aligned}$$

where $\mathbf{J}^\dagger \mathbf{J}$ is the projection operator, which exhibits zero divergence, so the above formula holds.

## B  ADDITIONAL EXPERIMENTAL SETTINGS

**Experimental Resources**   Our all experiments were performed on a cluster of 8 NVIDIA V100 or 8 H800 GPUs. Training time and resources do not increase significantly compared to the base model.

**Code**   Our code for experiments in this work is provided in supplementary materials and will be open-source upon acceptance.

**Used Augmentations**   Below are the data augmentation methods (with hyperparameters in parentheses) used by ScoreAug in Tables 2 and 3. During each training session, one augmentation method is randomly selected with equal probability across all types, where "identity" denotes unchanging the input samples. Notably, our approach operates effectively on both pixel and latent spaces, making it fully compatible with SiT models.

- **ScoreAug on EDM w/o NLA**
  - Unconditional CIFAR-10: brightness ($B = 2$), translation ($R_t = 0.25$), cutout ($R_c = 0.5$), rotation.
    - Conditional CIFAR-10: brightness ($B = 2$), translation ($R_t = 0.25$), cutout ($R_c = 0.5$), rotation.
    - FFHQ: translation ($R_t = 0.25$), cutout ($R_c = 0.5$), rotation.
    - AFHQ: translation ($R_t = 0.25$), cutout ($R_c = 0.5$), rotation.
- **ScoreAug on EDM w/ NLA**
  - Unconditional CIFAR-10: translation ($R_t = 0.125$), cutout ($R_c = 0.25$).
    - Conditional CIFAR-10: translation ($R_t = 0.125$), cutout ($R_c = 0.25$).
    - FFHQ: identity, translation ($R_t = 0.125$), cutout ($R_c = 0.25$).
    - AFHQ: brightness ($B = 2$), translation ($R_t = 0.125$), cutout ($R_c = 0.25$).
- **ScoreAug on SiT**, ImageNet: translation ($R_t = 0.0325$)

## C    ETHICS STATEMENT

By improving generalization of diffusion models, the method could broaden real-world applications like medical imaging and creative design where training data is scarce. However, mitigating memorization risks also helps prevent unintended data replication, supporting ethical AI development and copyright compliance in generated content.

## D    THE USE OF LARGE LANGUAGE MODELS

We use large language models (LLMs) for language editing, including copyediting, wording refinement, and minor stylistic polishing. LLMs does not contribute to idea generation, experimental design, code implementation, or result selection. All LLM edits are manually reviewed and validated.

## E    MORE QUALITATIVE RESULTS

Below are the qualitative results on CIFAR-10 (rotation controllable), AFHQ, and FFHQ.

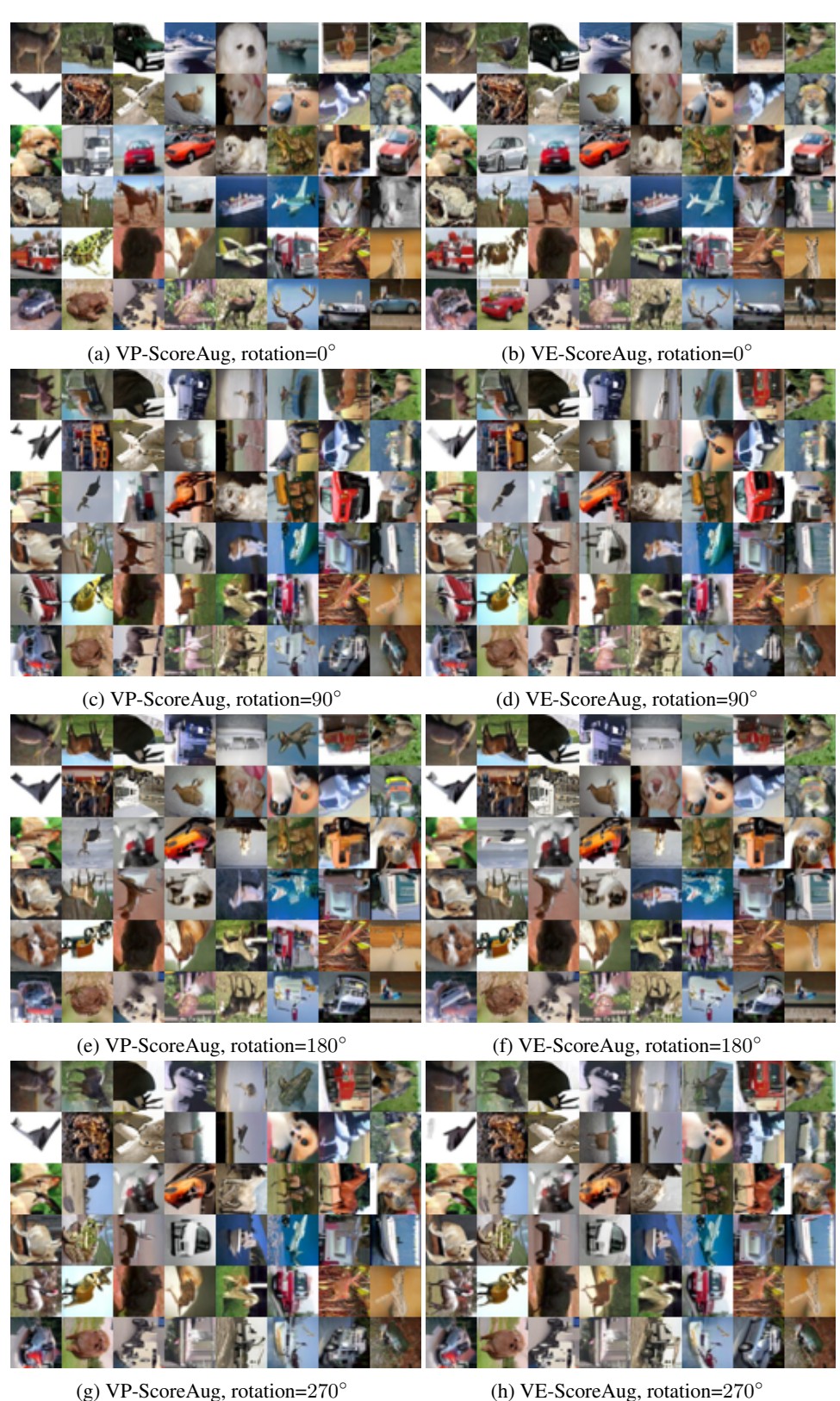

(a) VP-ScoreAug, rotation=0°    (b) VE-ScoreAug, rotation=0°

(c) VP-ScoreAug, rotation=90°    (d) VE-ScoreAug, rotation=90°

(e) VP-ScoreAug, rotation=180°    (f) VE-ScoreAug, rotation=180°

(g) VP-ScoreAug, rotation=270°    (h) VE-ScoreAug, rotation=270°

Figure 3: Augmentation-conditional generated images on unconditional CIFAR-10 of ScoreAug.

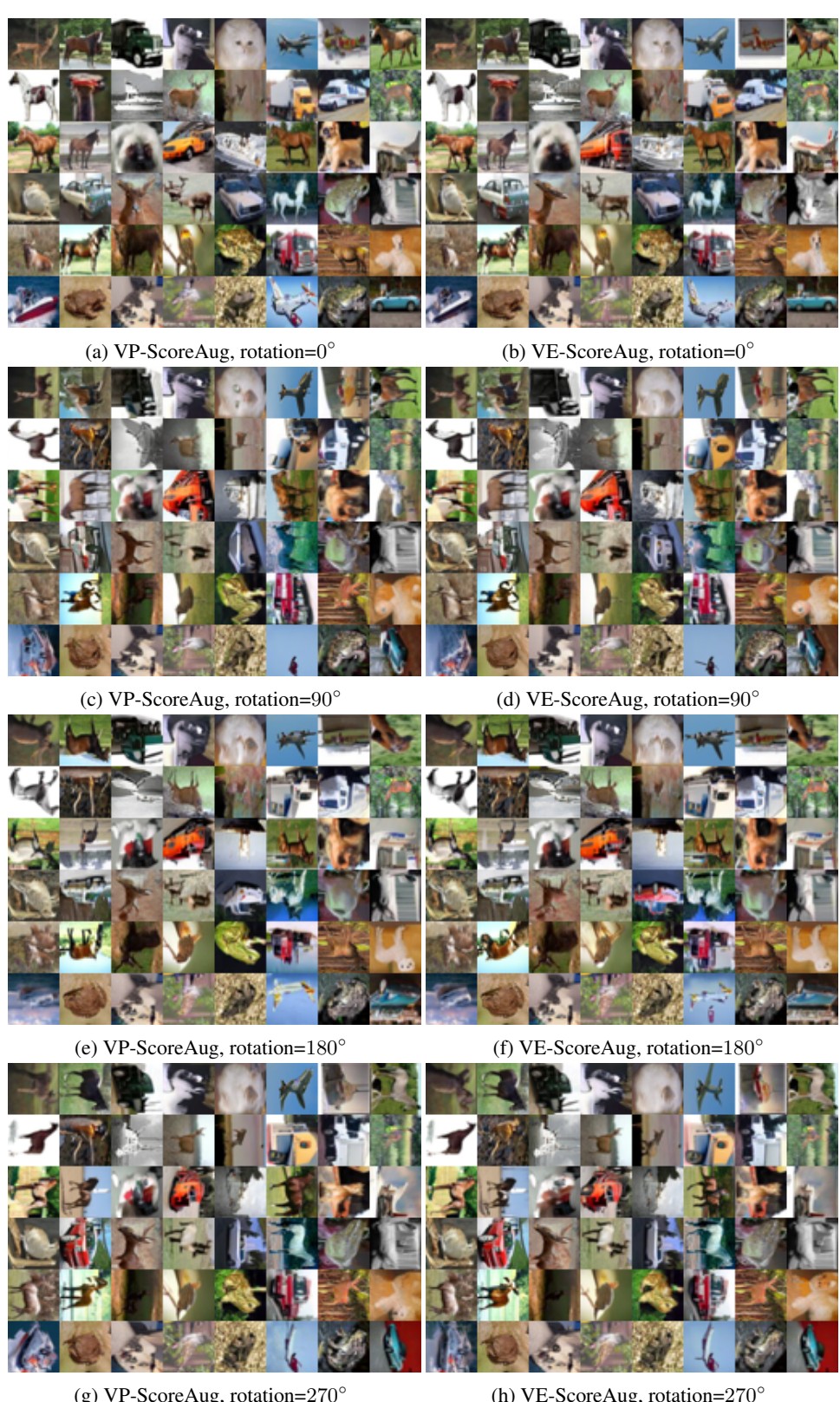

(a) VP-ScoreAug, rotation=0°      (b) VE-ScoreAug, rotation=0°

(c) VP-ScoreAug, rotation=90°      (d) VE-ScoreAug, rotation=90°

(e) VP-ScoreAug, rotation=180°      (f) VE-ScoreAug, rotation=180°

(g) VP-ScoreAug, rotation=270°      (h) VE-ScoreAug, rotation=270°

Figure 4: Augmentation-conditional generated images on conditional CIFAR-10 of ScoreAug.

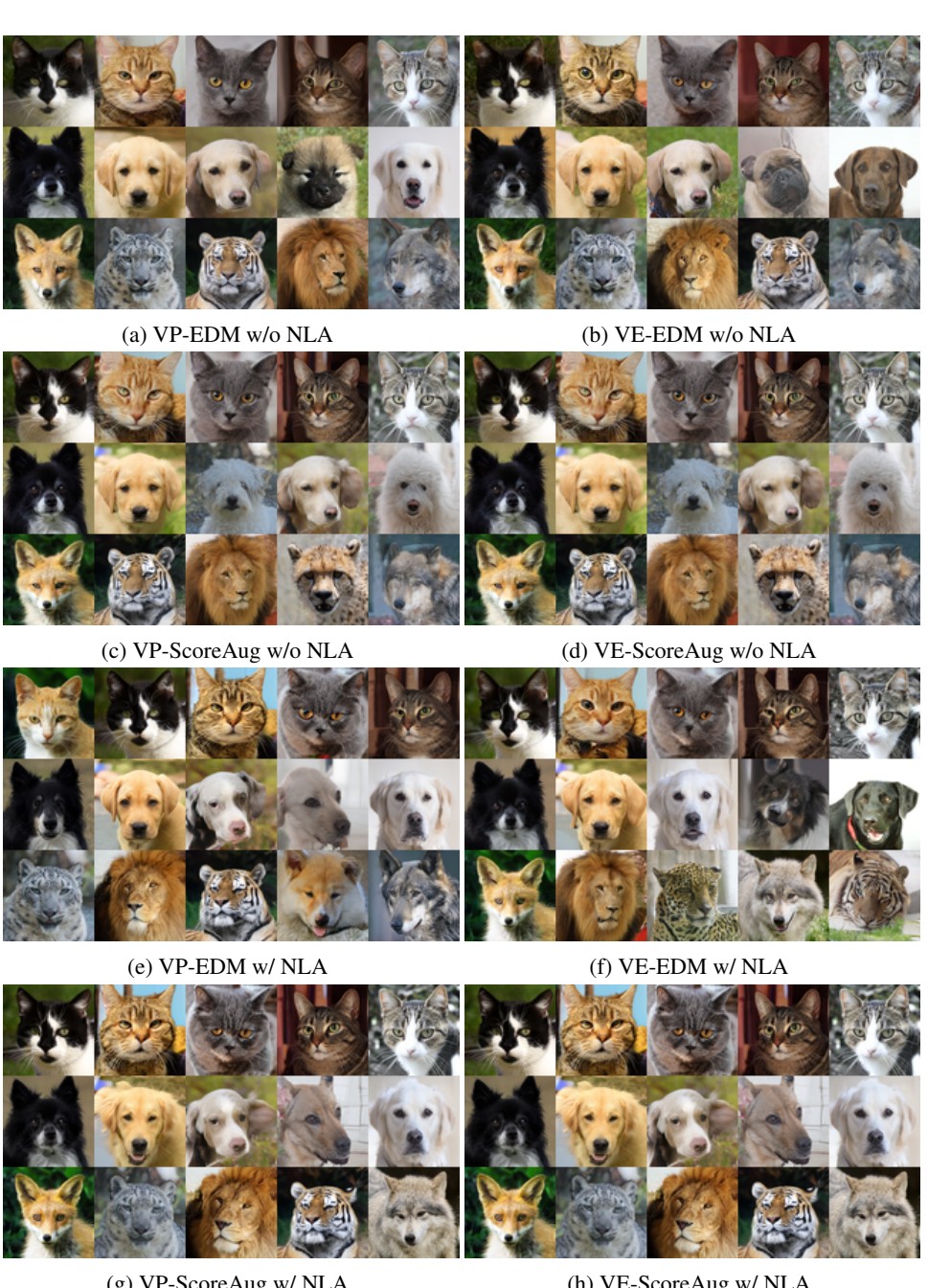

(a) VP-EDM w/o NLA         (b) VE-EDM w/o NLA

(c) VP-ScoreAug w/o NLA      (d) VE-ScoreAug w/o NLA

(e) VP-EDM w/ NLA         (f) VE-EDM w/ NLA

(g) VP-ScoreAug w/ NLA      (h) VE-ScoreAug w/ NLA

Figure 5: Generated images of EDM and ScoreAug without and with NLA on AFHQ.

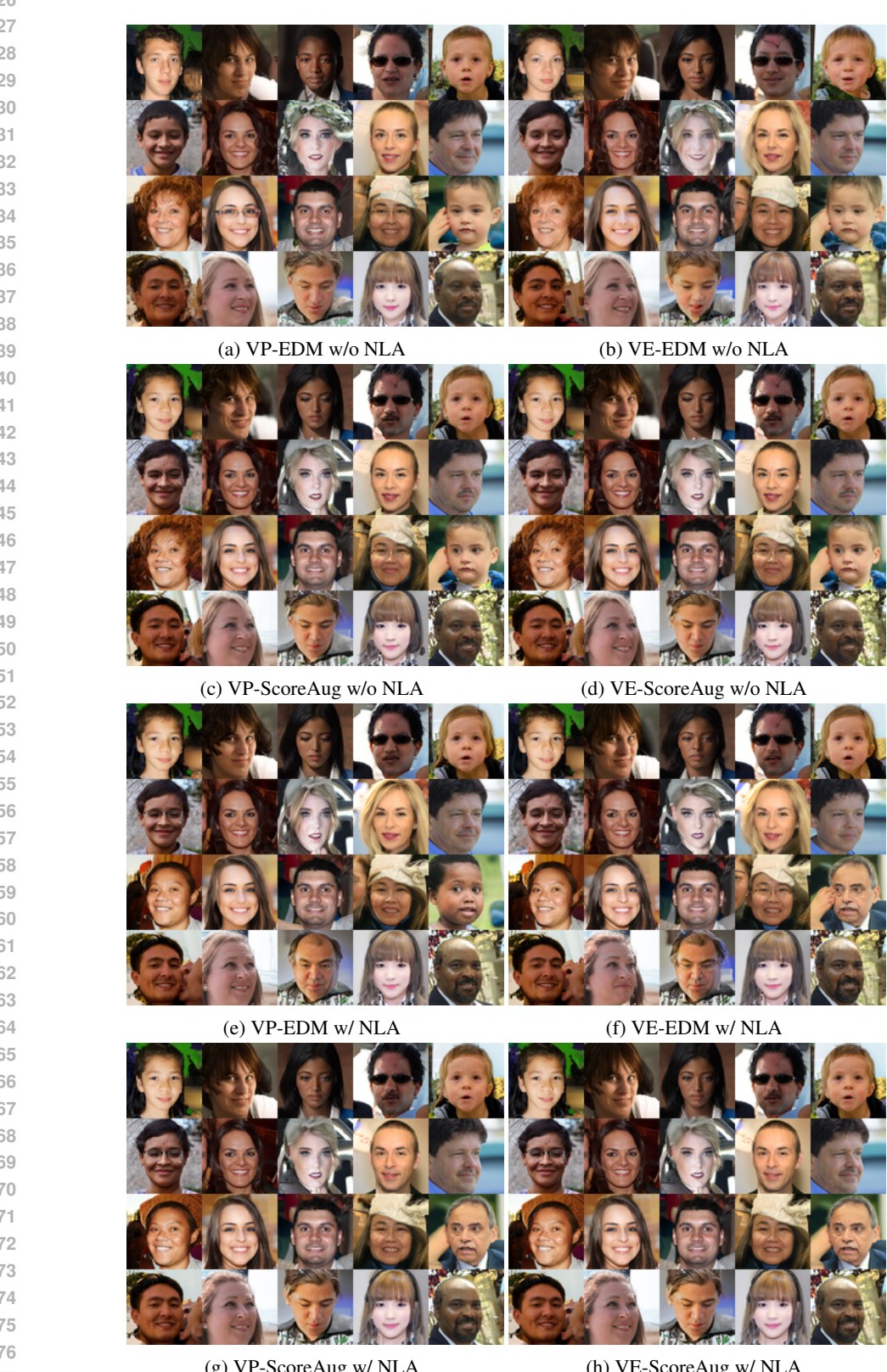

Figure 6: Generated images of EDM and ScoreAug without and with NLA on FFHQ.

