# OpenReview forum: "Score Augmentation for Diffusion Models"
_ICLR.cc/2026/Conference — ICLR 2026 Conference Withdrawn Submission_

### Official Review · Reviewer_Ko2W · 2025-10-28

**Soundness:** 3
**Presentation:** 3
**Contribution:** 3
**Rating:** 6
**Confidence:** 4

**Summary:**

The paper introduces Score Augmentation (ScoreAug), a diffusion-aligned augmentation framework that operates on noisy inputs and trains the denoiser to predict transformed clean targets, yielding an equivariant learning objective. The authors provide a score–transformation analysis (incl. a general theorem) and demonstrate gains across CIFAR-10, FFHQ, AFHQv2, and ImageNet with both U-Net/EDM and DiT/SiT backbones, showing better FID/sFID/IS, improved robustness to overfitting (small data, larger models), and more stable convergence.

**Strengths:**

1. Method–process alignment: Augmenting in the noisy space with an equivariant target is principled and avoids the mismatch of clean-only augmentation; it also mitigates augmentation leakage with proper conditioning.

2. A clear correspondence between scores under general transformations (Theorem 1) supports the design beyond linear cases.

3. Broad empirical coverage: Consistent improvements over EDM baselines (with/without non-leaky aug) and additional gains on SiT (ImageNet-256), indicating applicability beyond a single architecture/dataset.

4. Overfitting mitigation evidence: Stronger performance under reduced data, larger channel sizes, and throughout training (FID curves), matching the paper’s motivation.

**Weaknesses:**

1. Ablation depth: Type I vs. Type II under nonlinear transforms and the exact role/sensitivity of conditioning (and its injection method) deserve deeper, more granular analysis beyond a few tables.

2. Compute reporting: Claims that resources “do not increase significantly” are not quantified (e.g., wall-clock, GPU-hours) across datasets; clearer cost–benefit and sampling-time impacts would help adoption.

3. Metric/setting breadth: Heavy reliance on FID, with limited diversity metrics or human eval; text-conditional or higher-res benchmarks are only lightly touched (no large-scale T2I).

**Questions:**

see weakness

---

### Official Review · Reviewer_33yG · 2025-11-01

**Soundness:** 3
**Presentation:** 3
**Contribution:** 2
**Rating:** 2
**Confidence:** 4

**Summary:**

This paper introduces ScoreAug, a novel data augmentation framework specifically designed for training diffusion models. The goal is to alleviate overfitting and avoid artifacts caused by augmentation leakage.

The idea is to apply transformations to the noisy data rather than the clean data, and to train the denoiser to predict the transformed clean data rather than the original one, which corresponds to an equivariant learning objective. The authors provide a theoretical analysis that frames this approach as learning score functions in different transformed spaces.

Experiments on CIFAR-10, FFHQ, and ImageNet with U-Net and DiT architectures show that ScoreAug mitigates overfitting and improves generation quality.

**Strengths:**

- The paper aims to tackle an important and practical problem, providing an additional (and relatively understudied) perspective on improving the training of diffusion models.
- Figure 1 and Table 1 provide a clear and intuitive illustration of how ScoreAug works.
- The study presented in Figure 2 strongly supports the claim that ScoreAug effectively mitigates overfitting with limited data or excessive model capacity.

**Weaknesses:**

- The experimental setups described in Table 2 are very confusing, particularly the meaning of $+$ and $\times$.
  - Accoring to my understanding:
    - "EDM w/ NLA" = using a non-linear transformation pipeline (described in Table 6 of their paper), and conditioning the model with the 9-dim aug label vector $a$.
    - "ScoreAug(Linear)" = using one randomly selected linear transformation (Appx. B), and conditioning the model with the vector $\omega$ (Line 265).
    - "ScoreAug(type 1/2)" = using the same pipeline and conditioning as "EDM w/ NLA", but the input and target follow the definitions in Table 1.
  - Therefore, the "EDM w/ NLA + ScoreAug(Linear)" setup is particularly ambiguous. It is unclear how two distinct pipelines, conditioning vectors, and input-target definitions are combined to achieve a "synergistic" effect. Is it a sequential application, a random selection, or another strategy? Lines 805-806 indicate that the transformations differ from both ScoreAug(Linear)'s and EDM's.

- The experiments on the latent-space DiT (SiT) are preliminary.
  - The reported performance gain on SiT-XL is marginal, and results for smaller models like SiT-B are absent (smaller models can often show greater benefit).
  - Is the setup class-conditional or unconditional? Will these somewhat marginal improvements be diminished when classifier-free guidance (CFG) is applied?
  - The choice of augmentation for ImageNet appears overly simple. The appendix suggests that only translation was applied in the VAE's latent space. Such a simple transformation for a complex dataset may not fully demonstrate the potential of ScoreAug.
  - Although the ImageNet dataset is much larger than CIFAR and not as "data-limited" in the generative modeling context, data augmentation is still crucial for representation learning. Recent work has illustrated the close relationship between representation learning, generation quality, and training efficiency in diffusion models. Therefore, data augmentation on latent-space ImageNet and even larger-scale datasets may be very important. I was hoping the paper would provide an effective solution in this direction, but unfortunately, it seems to only introduce a weak "translation" with limited improvement.

- Presentation issues.
  - The reported FID scores are inconsistent in Table 2, 4, and 5.
    - Table 2 and Table 4 appear to be the same configuration for CIFAR-10, but the FIDs are different (2.35, 2.24, 2.11, 2.25 vs. 2.27, 2.29, 2.11, 2.06).
    - The result in Table 5 (2.21, 2.12, 2.01, 2.08, without brightness) is better than that in Table 4 (with all four augmentations).
  - Figure 1 and Table 1 illustrate the same idea and could be merged for simplicity.
  - Figure 2 and Table 5 are referenced in the text on Page 4, long before they actually appear.

**Questions:**

Please refer to the weaknesses regarding the "EDM w/ NLA + ScoreAug(Linear)" setup, and particularly the latent-space DiT experiments.

---

### Official Review · Reviewer_MqQy · 2025-11-03

**Soundness:** 3
**Presentation:** 3
**Contribution:** 1
**Rating:** 4
**Confidence:** 3

**Summary:**

This paper proposes ScoreAug, a data augmentation framework that applies transformations directly to noisy inputs in diffusion models, enforcing an equivariant objective that aligns augmentation with the denoising process. It claims improved generalization and stability across datasets (CIFAR-10, FFHQ, AFHQv2, ImageNet) with modest FID gains over EDM and EDM w/ NLA.

**Strengths:**

It's interesting to confirm that the transformation at the noisy space works as much as the original EDM training.

**Weaknesses:**

### W1. Marginal empirical gains.

- Table 2 already shows EDM w/ NLA ≈ 2.1 FID, while the proposed method achieves only ~0.05–0.1 improvement—well within run-to-run variance.
- The authors describe this as a “consistent performance improvement,” but such a small delta is unlikely to be statistically meaningful, particularly given the stochasticity of diffusion model training.
- Without repeated trials or confidence intervals, it is difficult to tell whether any real gain exists or if this reflects noise in evaluation. A more rigorous experimental protocol (multiple seeds, variance analysis) would be required to support the claimed benefit.

### W2. Limited conceptual novelty.

- The contribution would be stronger if the authors demonstrated clear failure modes of EDM w/ NLA and showed that ScoreAug resolves them in a measurable way (e.g., improved generalization under severe data scarcity).
- Furthermore, the relevance of augmentation for large-scale diffusion models is questionable: models trained on hundreds of millions of images already exhibit strong data diversity, so augmentation provides little marginal value.
- If the authors believe augmentation still matters, they should justify this by analyzing genuinely low-data domains (e.g., medical, scientific, or niche artistic datasets) where overfitting remains critical.

### W3. Overstated significance.

- The practical contribution seems incremental rather than a new paradigm for regularizing diffusion models. Clarifying the novelty boundary with EDM and reporting statistical significance would strengthen credibility.

**Questions:**

-

**Details Of Ethics Concerns:**

Not applied

---

### Note · Authors · 2025-11-13

I have read and agree with the venue's withdrawal policy on behalf of myself and my co-authors.